# A Robust and Opponent-Aware League Training Method for StarCraft II

**Ruozi Huang    Xipeng Wu    Hongsheng Yu    Zhong Fan**

**Haobo Fu**[*]    **Qiang Fu    Wei Yang**

Tencent AI Lab, Shenzhen, China
{rosiehuang,haroldwu,yannickyu,zhongfan,haobofu,leonfu,willyang}@tencent.com

## Abstract

It is extremely difficult to train a superhuman Artificial Intelligence (AI) for games of similar size to StarCraft II. AlphaStar is the first AI that beat human professionals in the full game of StarCraft II, using a league training framework that is inspired by a game-theoretic approach. In this paper, we improve AlphaStar's league training in two significant aspects. We train goal-conditioned exploiters, whose abilities of spotting weaknesses in the main agent and the entire league are greatly improved compared to the unconditioned exploiters in AlphaStar. In addition, we endow the agents in the league with the new ability of opponent modeling, which makes the agent more responsive to the opponent's real-time strategy. Based on these improvements, we train a better and superhuman AI with orders of magnitude less resources than AlphaStar (see Table 1 for a full comparison). Considering the iconic role of StarCraft II in game AI research, we believe our method and results on StarCraft II provide valuable design principles on how one would utilize the general league training framework for obtaining a least-exploitable strategy in various, large-scale, real-world games.

## 1  Introduction

As one of the most famous real-time strategy games, StarCraft II poses great challenges, in terms of developing a superhuman Artificial Intelligence (AI), to existing Reinforcement Learning (RL) techniques [Vinyals et al., 2017, 2019, Han et al., 2020, Wang et al., 2021], which mainly come in two-folds. The first challenge is the enormous space of strategies. There are approximately $10^{26}$ possible choices of actions at each time step, and each game normally consists of tens of thousands of time steps. The second challenge is the non-transitive strategy space with hidden information. Non-transitive strategies mean strategy A winning against B and strategy B winning against C does not necessarily mean strategy A winning against C. This implies simple self-play RL algorithms may exhibit strategic cycles Lanctot et al. [2017], Fu et al. [2021], making little progress in terms of the strength of the trained agent. For the hidden information, opponent entities are observable only when they are currently seen by the agent's own entities, and the information of opponent entities is extremely important in StarCraft II: it reveals the opponent strategy. Yet, it is worth-mentioning that the agent is allowed to actively demystify (known as scouting) the hidden information in StarCraft II, which is in contrast to other games with hidden information, such as poker.

As the first AI that defeated top humans in the full game of StarCraft II, AlphaStar [Vinyals et al., 2019] copes with the challenge of enormous strategy space by using human data. The RL agents in

---

[*]Corresponding author.

Table 1: A full comparison between AlphaStar and our method ROA-Star. The computational cost is for a single training agent in the league, and the evaluation results are based on Protoss vs. Protoss matches. ROA-Star uses significantly less computational resources than AlphaStar, and the evaluation against top humans is more comprehensive in ROA-Star than AlphaStar.

| | | AlphaStar | ROA-Star |
|---|---|---|---|
| computational cost | TPU or GPU | 256 3rd-generation TPU cores | 64 NVIDIA V100 GPUs |
| | CPU | 4100 preemptible CPU cores | 4600 standard CPU cores |
| | actor | 16000 concurrent games | 2400 concurrent games |
| | learner | 50000 steps per second | 11000 steps per second |
| human evaluation | amateur | 11 wins and 3 losses | 20 wins and 0 losses |
| | professional | 5 wins 0 losses (vs TLO [2]) 
 5 wins 1 losses (vs MaNa[3]) | 3 wins 2 losses (vs herO[3]) 
 10 wins 10 losses (vs Jieshi[3]) 
 12 wins 8 losses (vs Cyan[3]) 
 10 wins 10 losses (vs MacSed[3]) |

AlphaStar are all initialized by Supervised Learning (SL) on human replays. Also, a KL-divergence (from human replays) regularization is imposed during RL in AlphaStar. To deal with the non-transitive strategy space with hidden information, AlphaStar employs a league training framework that is inspired by game-theoretic approaches [Heinrich et al., 2015, Lanctot et al., 2017] with convergence guarantees to Nash Equilibrium (NE), which is an optimal solution concept AlphaStar aims for. The AlphaStar league consists of four (yet three types) constantly-learning agents: one main agent, one main exploiter, and two league exploiters. The main agent is the one output for evaluation after training. The main exploiter aims to find weaknesses in the main agent, while the league exploiters aim to find weaknesses in the entire league.

Despite achieving a Grandmaster level on Battle.net and defeating several top humans, AlphaStar is computationally prohibitive (see Table 1), and its main agent was later found fragile to some uncommon counter-strategies [4]. This implies the inefficiency of the AlphaStar league training framework in terms of approximating a NE strategy in large-scale games such as StarCraft II. In this paper, we improve the AlphaStar league training framework in two substantial ways. We found that (as shown in Figure 6), in the later training iterations of AlphaStar, the exploiters tend to lose the ability to identify the weaknesses in the main agent and the entire league. To alleviate this problem, we train goal-conditioned (as opposed to unconditioned exploiters in AlphaStar) exploiters that exploit the weaknesses in certain directions. To alleviate the problem that AlphaStar does not respond to the opponent real-time strategy effectively, we introduce a novel opponent modeling auxiliary training task, which conveys an explicit training signal for the feature extraction process to focus on the area of observation that reveals the opponent strategy most. In addition, we construct a novel "scouting" reward based on the prediction of the opponent strategy, which greatly encourages the scouting behaviors and thus helps the agent respond to the opponent real-time strategy more effectively. We term our new method as robust and opponent-aware learning training method for StarCraft II (ROA-Star).

We validate our improvements by comparing ROA-Star to AlphaStar. Extensive experiments demonstrate that the exploiters in ROA-Star are more effective in detecting the weaknesses of the main agent and the entire league; that the main agent in ROA-Star responds to the opponent strategy more effectively; and overall that the main agent in ROA-Star is significantly stronger. We also conducted by far the most comprehensive AI vs. top human evaluations in StarCraft II, where our agent trained by ROA-Star achieved a winning rate above $50\%$ in repeated games. A detailed comparison, in terms of the computational cost and human evaluation, between AlphaStar and ROA-Star is given in Table 1. In light of the significant improvement of ROA-Star over AlphaStar, we believe ROA-Star provides two insightful design principles, i.e., the goal-conditioned exploiters and the opponent modeling auxiliary task, on how one would utilize a league style training framework for obtaining a least-exploitable strategy in various, large-scale, real-world games.

---

[2]TLO is a professional player majored in Zerg not Protoss.

[3]The rankings of herO, MaNa, Jieshi, Cyan and MacSed among all professional players of Protoss are 2, 13, 19, 25 and 39 according to http://aligulac.com/periods/343/?page=1&race=p&nats=all&sort=vp

[4]In their released human evaluation replays (https://www.nature.com/articles/s41586-019-1724-z#Sec32), AlphaStar was easily defeated by Grandmaster players using some uncommon strategies, such as Cannon Rush.

## 2 Preliminary on StarCraft II and AlphaStar

### 2.1 StarCraft II

In StarCraft II [Vinyals et al., 2017], players act as battlefield commanders and start with a few workers who can gather resources. As resources accumulate, players need to allocate them to build buildings, train military units, and research new technologies. StarCraft II offers players the choice of dozens of unit types, each with different spells and abilities. It requires players to construct their armies strategically and assign different real-time tasks to these military units, which can be roughly divided into scout, defense, and attack. When the offensive units of two players meet, each player must quickly direct their units to engage in combat and effectively control them. To win the game, players require to control units at the micro level while making strategic choices at the macro level. Another fundamental setting in StarCraft II is the "fog of war", which limits a player's vision of the map to only those areas that are within the visual range of their entities (buildings and armies).

### 2.2 Supervised Learning in AlphaStar

Each agent in AlphaStar is firstly trained through supervised learning on human replays. Formally, an agent's policy is denoted by $\pi(a_t|o_{1:t}, a_{1:t-1}, z)$, where $\pi$ is represented by a deep neural network. At each time step $t$, the agent receives a partial observation $o_t$ of the complete game state $s_t$ and selects an action $a_t$. $o_t$ consists of its own entities, visible enemy entities, and a mini-map depicting the terrain of the battlefield. $a_t$ includes the choice of action type, the action executors, and the targets. The policy conditions on an extra statistic $z$, which is a vectorized description of a strategy. $z$ encodes the first 20 build order (buildings or units) and some cumulative statistics present in a game. In supervised learning, AlphaStar minimizes the KL divergence between human actions and the conditioned policy $\pi$ on human data.

### 2.3 RL and League Training in AlphaStar

To address the game-theoretic challenges, AlphaStar proposes a multi-agent RL algorithm, named league training. It assigns the learning agents three distinct types (main agent, main exploiters, and league exploiters), each corresponding to different training and opponent selection mechanisms. All agents are initialized with the parameters of the supervised learning agents and trained with RL. During the league training, these agents periodically add their checkpoints into the league.

The Main Agent (MA) takes the whole league as opponents, with a Prioritized Fictitious Self-Play (PFSP) mechanism that selects the opponent for RL training with probabilities in proportion to the opponent's win rate against MA. Also, the MA is trained by randomly conditioned on either zero or a statistic $z$ sampled from $\mathcal{D}$, where $\mathcal{D} = \{z_1, z_2, \ldots, z_m\}$ is a set of strategy statistics extracted from human data. Similar to goal-conditioned RL [Schaul et al., 2015, Veeriah et al., 2018], the agent, when conditioned on $z$, would receive pseudo-rewards that measure the edit distance between the target build orders in $z$ and the executed ones.

There are two types of exploiters that are designed to identify the weaknesses of their opponents. The Main Exploiter (ME) trains against the MA. The League Exploiter (LE) trains against the whole league. Both exploiters are unconditionally trained. Exploiters add a checkpoint to the league when they achieve a sufficiently high win rate or reach a timeout threshold. At that point, there is a certain probability that exploiters are reset to the supervised agent.

## 3 ROA-Star

In this section, we present the technique details of ROA-Star. As we discussed before, the purpose of the exploiters in AlphaStar is to find weaknesses of the MA and the entire league. Yet, as we found in the experiments, exploiters in AlphaStar (we re-implement AlphaStar ourselves) gradually lose the ability to counter the MA or the entire league as the training proceeds. This may be because it is increasingly difficult for the exploiters to counter strong agents in later training iterations via exploring the whole strategy space freely (other than initialized from the SL agents). Hence, we propose to alleviate this problem by training exploiters that exploit in certain directions with goal-conditioned RL. In real-time games like StarCraft II, the opponent's strategy, involving both the

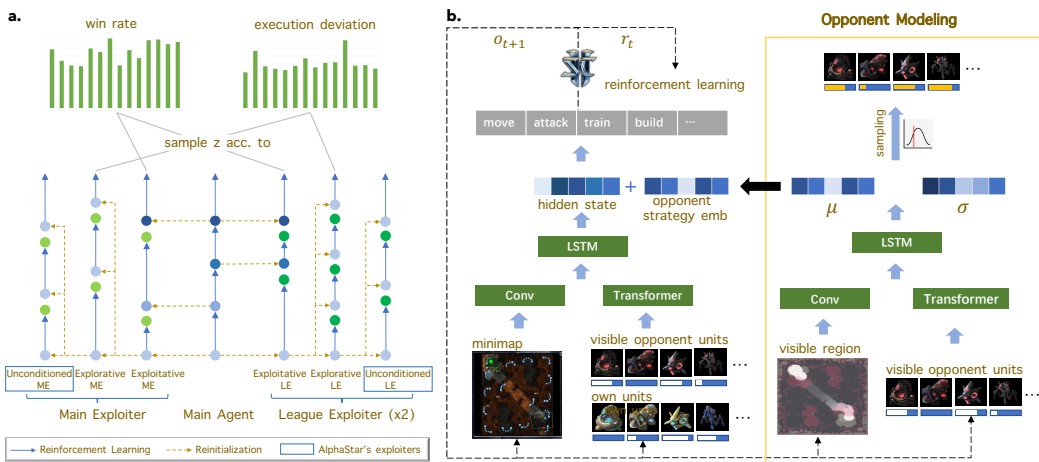

Figure 1: **a.** The design of goal-conditioned exploiters in ROA-Star. Blue rectangles highlight the original unconditioned exploiters in AlphaStar. Small circles are the agent models. Green histograms display the win rate and the execution deviation of $z$ in MA during a specific time slice in training. **b.** Opponent Modeling in ROA-Star. Left is the overview of the architecture of the agent in ROA-Star, which combine the embedding of opponent strategy. Right is the overview of the opponent prediction model.

composition of the entity and the technological development, can change rapidly. AlphaStar exhibits a somewhat slow response to these dynamic changes, resulting in the decrease in the playing strength. Our straightforward solution is introducing an opponent modeling training task. Also, based on the opponent modeling prediction, we construct a novel "scouting" reward to encourage efficient scouting behaviors, which helps respond to the opponent real-time strategy more effectively.

### 3.1 Goal-conditioned RL for the Exploiters

The exploiters (either the main exploiter or the league exploiter) in AlphaStar are trained unconditionally, i.e., the policy $\pi(a_t|o_{1:t}, a_{1:t-1})$ depending on only previous observations and actions. In addition to the unconditioned exploiters in AlphaStar, we introduce another two ways of training the exploiters $\pi(a_t|o_{1:t}, a_{1:t-1}, z)$ that are extra conditioned on the statistic $z$. More specifically, we train Exploitative Exploiters (EIE) that are conditioned on those $z$, which are associated with high win-rate in the MA. In other words, EIE try to find the weaknesses of either the main or the league based on "good" $z$ known so far. Also, we train Explorative Exploiters (ERE) that are conditioned on those $z$, which are under-explored so far in the MA. In other words, ERE try to find the weaknesses of either the main or the league based on "under-explored" $z$ so far. Both the EIE or ERE are trained with goal-conditioned RL, the details of which are described below.

The MA in ROA-Star is trained the same way as that in AlphaStar, where the MA is trained either unconditionally or conditioned on a statistic $z$ randomly selected from the set $\mathcal{D}$. In our case, we maintain a moving average of win rate of the MA for each sampled $z$ during the training process. This gives us an estimation of the "performance" of different $z$. Yet, it is worth mentioning that a MA conditioned on a certain $z$ is not guaranteed to generate plays that are consistent with the corresponding $z$. In other words, a $z$ with bad performance could be due to the fact that the MA is unable to execute the $z$ successfully. For this reason, we calculate the edit distance between the actual executed $z$ and the target $z$ as the execution deviation, and we maintain a moving average of the execution deviation for each $z$ in $\mathcal{D}$ as well.

We only sample from those $z$ with the Top-N win rate of the MA for EIE to learn. Similar to the training of MA in AlphaStar, EIE are rewarded by the $z$-related pseudo-reward measuring the edit distance between the target and executed $z$. Different from the exploiters in AlphaStar, EIE are always reset to the latest MA parameters whenever a reset occurs. As a result, EIE inherit the strongest strategies in MA. Also, once a $z$ is sampled, it is fixed till the next reset of EIE, which is in contrast to the MA training setting that samples $z$ for each game. Instead of evenly distributing computing

resources among all $z$ with the Top-N win rate, training with a fixed $z$ for each reset allows EIE to concentrate all its computational resources to refine the micro-level operations on that $z$ and uncover potential weaknesses at the micro level.

In comparison to EIE, ERE are conditioned on those $z$ that are currently under-explored according to the moving average execution deviation statistics of MA. We initialize ERE with the SL agents when reset occurs. We notice that the z-related pseudo-reward measured by the edit distance would be easily hacked [Ecoffet et al., 2021] on those $z$ with high execution deviation. For instance, the agent tends to produce the common military entities in target $z$ to gain reward but skip the critical ones. To reduce the execution deviation on these $z$, we design a reward that is similar to the trajectory-conditioned learning algorithm [Guo et al., 2020], which formulates a curriculum on how to imitate the target $z$. More specifically, ERE are rewarded for each correctly built entity in the target $z$ till the first mistaken entity happens, where we ignore the left entities in the target $z$. For the first mistaken entity (say $e_m$) in the target $z$, we additionally apply a strategy-guided loss:

$$\mathcal{L}_{sg} = -\sum_{i=1}^{d} e_m^{(i)} \log \pi(a_t^{(i)}|o_{1:t}, a_{1:t-1}, z),$$

where $e_m \in \{0,1\}^d, a_t \in \mathbb{R}^d$. $d$ is the dimension of action space, which encompasses all the actions related to entity construction. The strategy-guided loss is disabled for production-irrelevant actions, therefore it won't harm the learning of micro-operation.

We use both the curricular reward and strategy-guided loss in ERE training, which is similar to the idea of Go-Explore [Ecoffet et al., 2021]: the agent can always return to the promising state with the curricular reward and then explore the intended direction guided by the strategy-guided loss. Again, we only sample from those z with the Top-N execution deviation of the MA. Once a $z$ is sampled, it is fixed till the next reset of ERE. Also, as ERE are conditioned on the $z$ with high execution deviation, ERE aim to uncover potential weaknesses at the strategy (macro) level. An illustration of our exploiters in comparison to AlphaStar is given in Figure 1(a).

## 3.2 Opponent Modeling for the Entire League

Knowing the opponent strategy gives the agent a huge advantage in Starcraft II. Yet, with the basic setting "fog of war" in StarCraft II, it is difficult for agents to predict the opponent strategy. Nonetheless, it is possible to infer the opponent strategy to some extent based on the observed opponent entities. More importantly, because of the "scouting" mechanism in StarCraft II, the agent can actively send units to gather more information about the opponent. AlphaStar does not explicitly predict the opponent strategy, and there has been evidence [5] showing that AlphaStar does not respond to the opponent real-time strategy effectively, which greatly affect the agent's performance as demonstrated in our experiments.

In order to improve an agent's ability of responding to the opponent strategy promptly and effectively, we introduce an opponent modeling auxiliary task to infer the opponent strategy. In particular, during the supervised learning of human data, we train a probabilistic encoder to obtain a latent opponent strategy embedding $h$, which serves as input to a probabilistic decoder that explicitly predicts the opponent strategy. The encoder and the decoder are trained by maximizing the Evidence Lower Bound (ELBO) used in $\beta$-Variational Autoencoders ($\beta$-VAE) [Kingma and Welling, 2014, Higgins et al., 2014]. Afterwards, in our league training, the probabilistic encoder is fixed and serves as a special feature extractor that focuses the parts of observation that mostly reflects the opponent strategy.

To construct the input features for the encoder $q_\phi(h_t|o_{\leq t})$ of our opponent modeling task, we filter out opponent irrelevant information (e.g., own entities) in $o_t$ and focus on visible opponent information, which includes opponent armies, buildings, and technologies. The encoder $q_\phi(h_t|o_{\leq t})$ predicts a Gaussian distribution of $h_t$ with mean $\mu_t$ and variance $\sigma_t$. A KL divergence term is optimized between the predicted Gaussian and the standard Normal distribution. The decoder $p_\theta(y_t|h_t)$ predicts (a classification problem) the opponent invisible information at each time step, which includes the

---

[5]From their released human evaluation replays (https://www.nature.com/articles/s41586-019-1724-z#Sec32), it's relatively easy to observe that AlphaStar rarely conducts effective scouting and lacks knowledge of the opponent's real-time strategy, making it fragile to uncommon strategies like the Cannon Rush strategy in Protoss vs. Protoss matches.

quantity of each entity, the technologies being researched, and the location of important tech buildings (e.g., near the player's base or near the opponent). To summarize, we use the reparametrization trick [Kingma and Welling, 2014] to optimize the following loss:

$$\mathcal{L}_{om}(y_t, o_0, \ldots, o_t; \theta, \phi)$$
$$= -\mathbb{E}_{q_\phi(h_t|o_{\leq t})}[\log p_\theta(y_t|h_t)] + \beta KL(q_\phi(h_t|o_{\leq t})\|\mathcal{N}(0,1)),$$

where in practice the negative log likelihood in the left term is replaced with the focal loss [Lin et al., 2017] in our case to alleviate the class imbalance problem in StarCraft II. An illustration of the opponent prediction model is given in Figure 1(b).

In addition, we construct a novel "scouting" reward based on the change in the cross-entropy of the opponent strategy prediction. Other than predicting the opponent strategy behind the fog, another way to "defog" is to drive military units to periodically scout the area outside the vision. Given that the full map is too vast to scout entirely, it is crucial for the agent to determine when and where to scout. As a result, good "scouting" behavior should improve the prediction accuracy of the opponent strategy. We thus use the opponent prediction model to obtain such a "scouting" reward. At each step of the RL process, we calculate the cross-entropy between the true opponent strategy and the predicted probabilities of the classifiers in the opponent prediction model:

$$H(t) = -\sum_{i=1}^{N} y_t^{(i)} \log P_t^{(i)}$$

We reward the agent every 30 seconds with the decrease in cross-entropy to encourage scouting behaviors. Note that the agent will not be punished for the increase in cross-entropy, as there is a sustained growth of cross-entropy as the game progresses (prediction in later stage is much harder).

$$r_{scout}^t = \max(H(t) - H(t^{min}), 0),$$
$$where\ t^{min} = \arg\min_{t'} H(t'),\ t' \in (t, t+30s).$$

As demonstrated in our experiments, the "scouting" reward greatly encourages the effective "scouting" behavior and improves the overall performance.

## 4    Related Work

There is enormous literature on either goal-conditioned RL [Andrychowicz et al., 2017, Florensa et al., 2018, Ghosh et al., 2019, Chane-Sane et al., 2021, Liu et al., 2022a] or opponent modeling [He et al., 2016, Albrecht and Stone, 2018, Zheng et al., 2018, Raileanu et al., 2018, Willi et al., 2022, Fu et al., 2022]. As in this paper we are focused on how goal-conditioned RL and opponent modeling would improve the league training efficiency of AlphaStar, we will describe related work only in the realm of AI for StarCraft, with more emphasis on literature that comes after AlphaStar.

Before AlphaStar, early research on StarCraft focus on traditional methods, such as planning methods associated with heuristic rules [Weber, 2010, Ontanón et al., 2013, Ontañón et al., 2008, Weber et al., 2011, Ontanon et al., 2013]. Later, there have been RL methods for micro-management in mini-games [Usunier et al., 2016, Vinyals et al., 2017, Du et al., 2019, Han et al., 2019]. Most recently, RL were applied to the full game with the help of some handcraft rules [Lee et al., 2018, Sun et al., 2018, Pang et al., 2019]. However, none of these works achieves a competitive human level in the full game of StarCraft II.

AlphaStar [Vinyals et al., 2019] became the first StarCraft II AI that achieves the GrandMaster level performance. After that, to enhance the strategy diversity in the league of AlphaStar, TStarBot-X [Han et al., 2020] leverages various human replays to create diverse initialization checkpoints for the exploiters to reset. Handcraft rules were also employed to help the MA explore the strategy space. Another work after AlphaStar is StarCraft Commander (SCC) [Wang et al., 2021], which makes great efforts to optimize the training efficiency by filtering the human data used in imitation learning and compressing the network structure. They branch off new main agents learning specified strategies. SCC displayed comparable performance to professional players in human evaluations, but its APM surpasses humans by a large margin due to the lack of APM limits. Recently, A hierarchical RL method has been proposed to train a StarCraft II AI that can defeat the built-in AI using very few computational resources [Liu et al., 2022b]. To summarize, none of these StarCraft II AIs is able to achieve the level of professional players while adhering to the APM limits and utilizing fewer computational resources than AlphaStar.

# 5  Experiments

After 50 days of league training, ROA-star generates a total of 768 models, of which 221 are MA models. We evaluate the robustness of the 50-day MA model under the condition of no $z$ through human evaluation. As a comparison, we replicated AlphaStar and trained it for 10 days. We compared the first 10 days of training of ROA-Star and AlphaStar from multiple dimensions. As ROA-Star contains improvement in both exploiters' training method and opponent modeling, we conduct the ablation experiments to demonstrate the effectiveness of each component. Each ablation experiment was trained for 5 days. Notably, all the comparative experiment and ablation experiments use the same computational resources as ROA-Star, as shown in the Appendix A.5.

## 5.1  The Effectiveness of Goal-conditioned Exploiters in ROA-Star

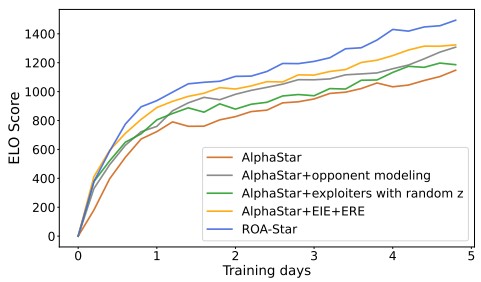

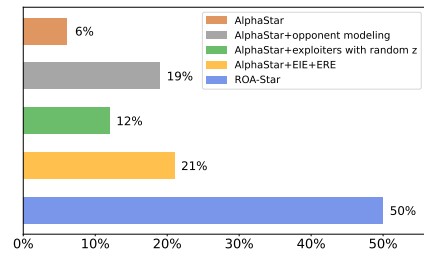

Figure 2: Comparison of each composition of ROA-Star using Elo score.

Figure 3: Worst-case win rate of the 5-day MA models

Based on the foundation experimental setup of AlphaStar, we modify the original unconditioned exploiters and design the following two experiments to test the effect of goal-conditioned exploiters.

**AlphaStar+exploiters with random z**: We make exploiters learn $z$ in $\mathcal{D}$ with uniform probability. For each reset, the exploiter randomly selects one $z$ from $\mathcal{D}$ to learn with an 80% probability and learns unconditionally with a 20% probability. When learning a selected strategy, the exploiter would be rewarded by the $z$-related pseudo-rewards measuring edit distance between the target and executed build orders.

**AlphaStar+EIE+ERE**: On the basic setting of AlphaStar, we utilize our goal-conditioned exploiters in Section 3.1, the details of the experimental setup are in the Appendix A.3.

To examine the relative strength of the MA models in different ablation experiments, we consider the rating metric Elo score. Elo scores reflect the relative skill levels of the players based on the win rates between them. We play 100 matches between any two of the first 5-day models of each MA and plot the Elo curves in Figure 2. Besides, we calculate the worst-case win rate of each 5-day MA model defending against all other MA models as a metric of robustness, the result is shown in Figure 3. As we can see, the exploiters that learn random strategies contribute to the robustness of MA, but our goal-conditioned exploiters obviously outperform random selection on $\mathcal{D}$.

Additionally, we demonstrate the capability of ERE to learn the precise build orders of target strategies through the use of curricular reward and strategy-guided loss. Our supplementary video and Appendix C.2 showcase the remarkable performance of ERE in imitating non-mainstream strategies.

## 5.2  The Effectiveness of Opponent Modeling in ROA-Star

Table 2: Win rate against different opponents.

| model win rate vs | AlphaStar | +opponent modeling |
|---|---|---|
| Void Ray push | 54% | **60%** |
| Proxy VD | 50% | **56%** |
| VC Stalker Play | 22% | **52%** |
| Disruptor push | 48% | **58%** |

Table 3: Comparison with AlphaStar

|  | AlphaStar | ROA-Star |
|---|---|---|
| worst win rate | 1% | **44%** |
| avg win rate | 70.2% | **84.1%** |
| RPP score | 0.2157 | **0.7843** |

We conduct an ablation study to validate the effectiveness of opponent modeling.

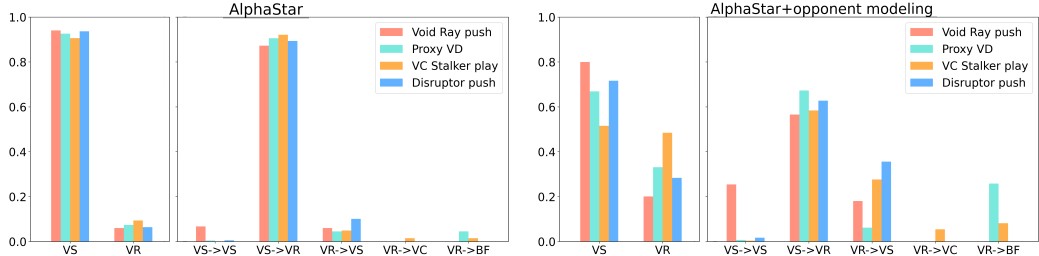

Figure 4: Distribution of tech-building routes against specific opponents. Building acronyms: VR (Robotics Facility), VS (Stargate), BF (Forge), VC (Twilight Council), VD(Dark Shrine).

**AlphaStar+opponent modeling**: With the aid of the opponent prediction model, we incorporate opponent strategy embedding for the original AlphaStar agents, as well as add scouting rewards. The worst-case robustness is in Figure 3 and the Elo curve is in Figure 2.

To better demonstrate the impact of opponent modeling, we test two 5-day MA models, one trained with opponent modeling and one without, against opponents with specific strategies. We generated these opponents by training MA only on the fixed $z$ we selected, without using exploiters. Figure 4 displays the distribution of tech-building routes when the models encounter various opponents. Although both models employ a similar dominant strategy, namely VS-VR, the model trained with opponent modeling demonstrates greater flexibility and has prompt responses even on its first tech-building when encountering different opponents, thereby exhibiting superior robustness compared to the original AlphaStar. For example, it increases the probability of VR opening to defend against the Blink Stalkers and build more BF and Photon Cannon to defend the cloaked Dark Templars. The models' win rates against these opponents are shown in Table 2.

In the Appendix C.6, we present visualizations of the latent space of the opponent prediction model, showcasing ROA-Star's "awareness" of the opponent's strategy when facing various opponents. We also compare the time consumed for the agent to discover the newly-built buildings in Appendix C.3. Figure 12 shows that scout reward remarkably reduces the time to discover the opponents' building under the fog.

### 5.3 Overall Comparison between ROA-Star and AlphaStar

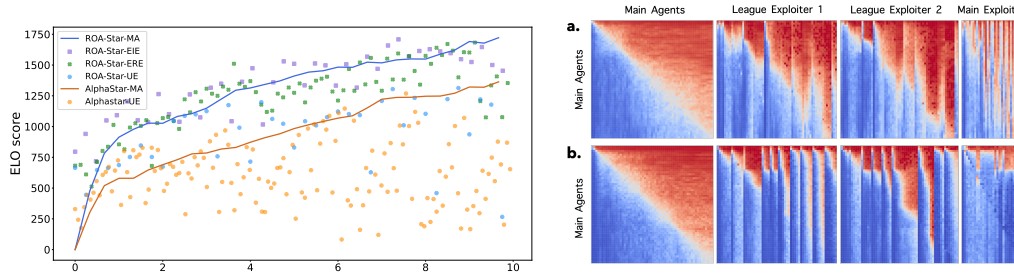

Figure 5: Elo scores of agents in the league of AlphaStar and ROA-star during 10-day training. Curves represent two main agents and points represent the models from exploiters. UE refers to unconditioned exploiters.

Figure 6: Payoff matrix calculated in (a). our ROA-Star and (b). AlphaStar. Blue means a row agent wins and red loses. The exploiters in AlphaStar lose efficacy in the later stage of training.

The primary objective of ROA-Star is the improvement in robustness. However, it's scarcely possible to precisely measure the robustness of the agent due to the enormous strategy space. Similar to the evaluation method in Section 5.1, we count the win rates by a Round Robin tournament on models of two leagues, with each pair of models playing 100 matches. Based on those matches, we introduce quantitative evaluation indexes to score the robustness of the MA and the whole league for AlphaStar and ROA-Star.

We calculate the win rate in the worst case and average case that two 10-day MA models defend against all models in two leagues. Results are shown in Table 3 that ROA-Star significantly outperforms AlphaStar in the MA's robustness. Relative population performance (RPP) is a measure of the expected outcome of the game between two populations. It is calculated based on their meta-game mixture strategies after they have reached Nash equilibrium [Balduzzi et al., 2019]. Given the payoff matrix $M_{AB} \in [0,1]^{N \times M}$ between all the models in league $A$ and league $B$ with two mixture strategies probabilities $P_A, P_B$ in the Nash equilibrium, the RPP of league A is:

$$RPP(A) = P_A^T M_{AB} P_B$$

As shown in Table 3, a higher RPP of ROA-Star demonstrates it constructs a mixture strategy that could counter any mixed strategy from AlphaStar.

As a by-product of the above matches, we evaluate the relative strength of all the players in the league of AlphaStar and our ROA-Star with the Elo rating system. The Elo score curves of the two main agents as well as points indicating the models from exploiters are plotted in Figure 5. As we can see, the strength of MA in ROA-Star is superior to AlphaStar throughout the training procedure, and ROA-Star can always generate more effective opponents for the MA.

Additionally, we evaluate the internal payoff estimation for each of the two leagues, giving the agents' relative performance against all the players in the same league. As shown in Figure 6, the exploiters of AlphaStar seem to have gradually weakened dominance to the MA in the later stage. Meanwhile, there always exist evenly matched opponents for MA in the ROA-Star league, which benefits from its goal-conditioned exploiters.

## 5.4   Top Human Evaluation of ROA-Star

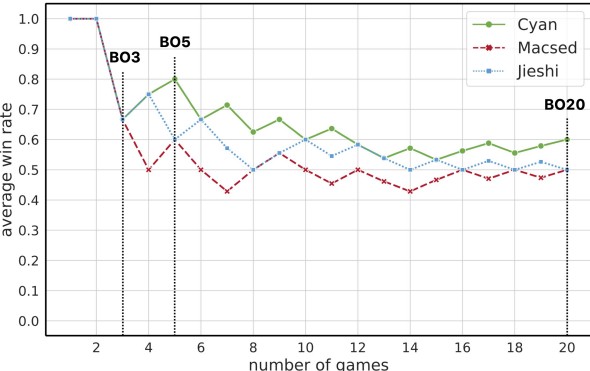

Figure 7: Trend of ROA-Star's win rate when fighting with professional players. ROA-Star won BO3 and BO5 matches against all the players and maintained final win rates no less than 50%.

To make a comprehensive evaluation of ROA-Star without access to the official Battle.net, we invite three top professional players: Cyan, Macsed and Jieshi to play against our agent on the Kairos Junction map. Although most of the current AI trained in the full games of StarCraft II are evaluated with human players in best-of-three (BO3) or best-of-five (BO5) matches [Vinyals et al., 2019, Wang et al., 2021], we realize that human has the ability to identify the weaknesses of an opponent in repeated competitions. Therefore, we ask each professional player to play 20 matches with ROA-Star to validate its robustness. We didn't make any demands on the strategies they use except encourage them to try more methods to find the weakness of our agent (Appendix B.1).

Figure 7 shows the trend of ROA-star's win rates as the games proceeds. ROA-star gains the upper hand over all the opponents at the beginning, but its win rate drops as the professional players learn more about it and test more strategies. Finally, as a mixed strategy, ROA-star maintains a win rate of no less than 50%, which means human players didn't find a winning strategy that can continuously defeat it, indicating the robustness of the agent.

We submit a comparison video with AlphaStar as supplementary material, showing how AlphaStar was defeated by the Cannon Rush strategy in Battle.net evaluation and how we defend against the same strategy played by the professional player through effective scout and prompt response.

## 6 Conclusions and Limitations

In this paper, we develop a new league training method, i.e., ROA-Star, for approximating a NE in very large-scale games such as StarCraft II. ROA-Star improves the influential AlphaStar league training in two significant aspects. ROA-Star trains goal-conditioned exploiters (as opposed to unconditioned exploiters in AlphaStar), which greatly improves the exploiters' ability in identifying weaknesses at both the micro and macro level for the MA and the entire league. In addition, a novel opponent modeling task is introduced in ROA-Star, which helps the agent respond to the opponent's real-time strategy more effectively. Meanwhile, a novel "scouting" reward is constructed based on the prediction of the opponent strategy. Extensive experiments demonstrate the effectiveness of ROA-Star over AlphaStar. Overall, ROA-Star produced a better and superhuman AI with orders of magnitude less resources than AlphaStar on StarCraft II.

There are several future directions of this work. During league training, we kept the parameters of the opponent prediction model fixed to retain the prior knowledge of game strategies in the human data. This approach proved effective in our experiments, yet it is worth investigating the impact of the diversity of strategies in the human data. Also, the policy diversity [Wu et al., 2022, Yao et al., 2023] during league training is worth pursuing as well. In this paper, we only tested ROA-Star on StarCraft II, and future experiments on other large scale games with hidden information are needed to further validate the effectiveness and generalization of ROA-Star. Finally, even though ROA-Star provides a general guideline that goal-conditioned exploiters and opponent modeling help in terms of approximating a NE in very large-scale games, recent research advances on both goal-conditioned RL and opponent modeling in general are definitely worth exploiting to further improve ROA-Star's performance.

## Acknowledgements

We thank all the human players who participated in this work and helped us conduct professional evaluations. We would like to express our gratitude to Tencent AI Arena (https://aiarena.tencent.com/) for providing the powerful computing capability that facilitated the training of ROA-Star.

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

# A  Experiment Setup

The training procedure of ROA-Star includes a supervised learning stage and a 50-day multi-agent reinforcement learning stage. We also implement an experiment with AlphaStar's setup for comparison and trains with the same computation resources. Due to the limitation of budget, the comparison experiment lasts for 10 days which is still a valid baseline as the main agents always get transitive improvement [Vinyals et al., 2019]. Besides, we conduct a few ablation experiments to evaluate the impact of each component in ROA-Star, which are enumerated in Appendix B.2. Each ablation experiment was trained for 5 days. All these experiments are applied in the race Protoss and all the training of reinforcement learning is restricted on the Kairos Junction map. In this section, we introduce the basic settings of ROA-Star.

## A.1  Human Dataset

Blizzard is releasing a large number of 1v1 replays played on the ladder. The instructions for how to download the replay files can be found at https://github.com/Blizzard/s2client-proto. We extracted a dataset from these replays which contains 120,938 Protoss vs. Protoss replays from StarCraft II versions 4.8.2 to 4.9.3. These replays were played by human players with MMR scores greater than 4100.

We utilize the dataset of human replays to learn a good initiation checkpoint for reinforcement learning. After 5 days of supervised learning, the model trained on the full game of StarCraft II can defeat the built-in elite AI with a win rate of 90%. We also train an opponent prediction model on this dataset. The opponent prediction model converges after half-day training, its performance on the test set will be exhibited in Appendix C.4.

Human replays are also used to construct the strategy set $\mathcal{D}$ for the league training. In order to select a set that can cover the effective strategies, we extract strategy statistic $z$ from each human replay in the dataset and cluster all $z$ using the edit distance between their build orders. We sample from each cluster equally to ensure the diversity of selected $z$. Finally, we obtain 193 different $z$ which constitute the strategy set $\mathcal{D}$.

## A.2  Reinforcement Learning

In the reinforcement learning stage, we reward the agents with the win-loss outcome, $z$-related pseudo-rewards and scouting reward. Similar to AlphaStar's configuration, the $z$-related pseudo-rewards measure the edit distance between executed and target build orders, as well as the Hamming distance between executed and target cumulative statistics on the units, buildings, and technologies. When agents condition on no extra $z$, we disable all $z$-related pseudo-rewards. It's worth noting that ERE replace the edit distance reward with the curricular reward on the build orders.

We apply RL techniques similar to those used in AlphaStar. To perform asynchronous and off-policy updates, we use V-trace algorithm [Espeholt et al., 2018], as well as the self-imitation algorithm (UPGO, Oh et al. 2018). We also apply a standard entropy regularization loss and a policy distillation loss distilling from the last reset target, i.e. the historical MA model for EIE, and the supervised model for other agents. We apply an additional strategy-guided loss for ERE to help learn under-explored strategies. The overall loss we used in the reinforcement learning stage is shown below.

$$\mathcal{L}_{RL} = \mathcal{L}_{V\text{-}trace} + \mathcal{L}_{UPGO} + \mathcal{L}_{entropy} + \mathcal{L}_{distill} + \mathcal{L}^*_{ERE}$$

## A.3  League Setting

ROA-Star consists of four simultaneously training agents in its league: one MA, one ME, and two LEs, where the exploiters are categorized into ME and LE by the different ways of getting opponents. The MA trains with strategy statistic $z$ sampled from our strategy set $\mathcal{D}$, and we set z to zero 10% of the time. A frozen copy of MA is added as a new player to the league with a period of every $2 \times 10^8$ steps. The LE fights with the whole league and adds a frozen copy into the league when it defeats all the players in the league with a win rate above 70% or reaches the timeout threshold of $2 \times 10^8$ steps. At this point, its parameters will be reset with a 25% probability. ME aims to find the weakness of MA, it adds the frozen copy in the league and reset parameters when defeating MA in more than 70% of games or after a timeout of $4 \times 10^8$ steps.

So far, the league setting is almost the same with AlphaStar. In ROA-Star, we reform both ME and LE to goal-conditioned exploiters. During the training, each exploiter will be reset to various configurations with a proportion of 20% origin unconditional exploiter, 30% EIE, and 50% ERE. EIE reset to the current MA model. It samples from the $z$ set with the top 10% win rate. ERE reset to the supervised model and condition on top 15% $z$ in execution deviation. All the MA and the exploiters combine the opponent strategy embedding into their observations and get rewards from the scouting behaviors.

### A.4 APM Limits

There exists a physical limit for human players on the actions per minute (APM) they can execute. To ensure fairness, we set limitations on the operating frequency for AI that the agent should successively execute actions with a minimum decision interval of 3 frames (around 130 milliseconds). The average APM of any of our final agents' models is less than 240 (with a peak APM below 800), which is close to the human players on Battle.net according to AlphaStar [Vinyals et al., 2019].

### A.5 Infrastructure

To scale up league training, we utilize a distributed learner-actor framework depicted in Figure 14. Actors are deployed on CPU machines to interact with the StarCraft II environments, perform action inference and generate training samples. Meanwhile, learners are deployed on GPU machines to update the model parameters using these samples. League Manager is distributed across both the GPU and CPU, maintains the win rates of all historical models, and allocates training tasks to learners and actors, including the agent's own model, the opponent's model, and $z$ of both sides.

For each agent, the full scale of computational resources contains 64 NVIDIA v100 GPUs and 4600 CPU cores. Each actor worker occupies two CPU cores. About 2400 StarCraft II environments are used simultaneously to provide training samples for an agent. An agent processes about 11000 environment steps per second.

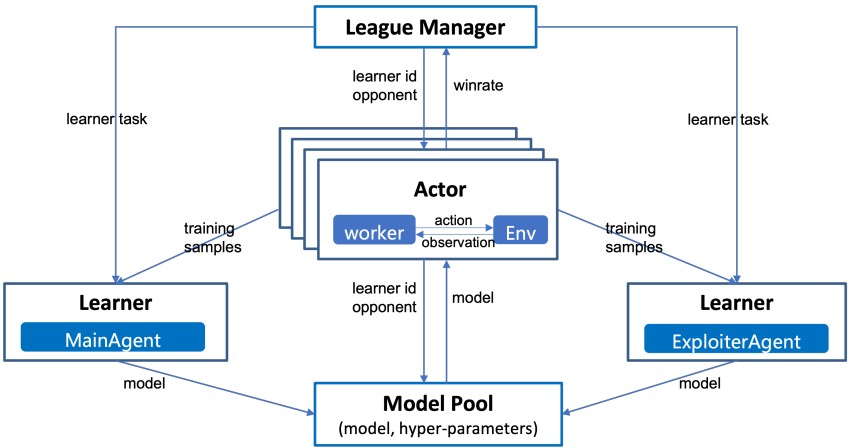

Figure 8: The framework of league training in ROA-Star.

### A.6 Summary of ROA-Star's Training Process

Here we present the summary of the 50-day training process of our agents, where the basic settings of each agent can be found in Appendix A.3. With the scale of our computational resources, an agent can process about 11,000 environment steps per second. Our MA was continuously trained for 50 days, which consumed about $4.42 * 10^{10}$ steps. During training, MA added a frozen copy to the league every $2 * 10^8$ steps, resulting in a total of 221 models. Our ME reset its parameters after $4 * 10^8$ steps at most, which generated 113 models in total. Our LE reset its parameters after $2 * 10^8$ steps at most, and the two concurrent-training LEs generated 216 and 218 models respectively.

In comparison, AlphaStar's MA was continuously trained for 44 days and consumed around $1.9 * 10^{11}$ steps (around 50,000 environment steps per second) [Vinyals et al., 2019].

# B   Evaluation Details

## B.1   Human Evaluation

To evaluate the robustness of ROA-Star, we invite three top professional players: Jieshi, Cyan and Macsed to play 20 matches each with ROA-Star. The matches take place on the Kairos Junction map, with both sides using the race Protoss. All of these professional players major in Protoss, and according to Aligulac[6], their world rankings in Protoss are 19, 25, and 39, respectively. They are champions of many StarCraft II professional competitions, including Dreamhack StarCraft II Masters China and StarCraft II World Championship Series China.

The 20 matches against every professional player were divided into 2 times, with an interval of one week in between. 10 matches were played at a time, with a 3-minute break between matches and a 20-minute break after 5 consecutive matches. Professional players could watch replays against ROA-Star and think about their strategy during each break. The average duration of each game was approximately 10 minutes, with the shortest being 3 minutes and the longest being 18 minutes.

To express our gratitude towards the professional players and motivate them to win, we offered them two options for calculating test fees before the test and allowed them to choose the one that suited them best. The first method is that the test fee for each match is 100 RMB, and the second method is based on the result of each match: professional players receive 150 RMB when they win the game, otherwise, they will only receive 50 RMB. Finally, Cyan chose the first method, while Macsed and Jieshi chose the second method.

In the end, we invited herO, the champion of DreamHack SC2 Masters 2022 Atlanta and the second-ranked professional player in Protoss according to Aligulac[6], to play two best-of-three (BO3) matches against our agent as the final benchmark. Prior to the competition, we made an agreement that herO would be rewarded with 100 dollars for every BO3 victory, and no payment would be made if he loses. In the end, we won the first BO3 with a score of 2:0, and lost the second BO3 with a score of 1:2, showing ROA-Star is competitive with the best human player in the world.

## B.2   Robustness Evaluation between AIs

It's hard to directly measure the models' robustness because of the vast space of cyclic, non-transitive strategies in StarCraft II. Instead, we apply the Round Robin tournament on the set of models to be evaluated, with any two models in the set playing 100 matches. Based on the performance of the models in these matches, we conducted robustness evaluations on different models and populations.

We conducted tournaments on two model sets. The first set includes the first 5 days' MA models of all the ablation experiments, including:

- **AlphaStar**: The original AlphaStar we replicated.
- **AlphaStar+exploiters with random z**: Reform the unconditional exploiters in AlphaStar to learn random strategies sampled from strategy set $D$.
- **AlphaStar+EIE+ERE**: Reform the unconditional exploiters in AlphaStar to EIE and ERE.
- **AlphaStar+ERE**: Reform the unconditional exploiters in AlphaStar to ERE.
- **AlphaStar+opponent modeling**: AlphaStar add opponent modeling.
- **Roa-Star**

We demonstrate the robustness evaluations on the first set in Section 5.1, Section 5.2, and Appendix C.1.

The second model set contains all the first 10 days' models in the league of AlphaStar and our ROA-Star, including the models generated by MA and exploiters. The robustness evaluations on the second set are shown in Section 5.3.

---

[6]http://aligulac.com/periods/343/?page=1&race=p&nats=all&sort=vp

# C Supplementary Experimental Results

## C.1 The Ablation Study about EIE and ERE

We conduct an ablation experiment to verify both EIE and ERE could contribute to the robustness of MA.

**AlphaStar+ERE**: During the training, we reset the exploiters to the configuration of ERE with a probability of 80%, and to the original unconditional exploiters with a probability of 20%.

We exhibit the Elo curves of each MA with different exploiters in Figure 9, ERE is superior to random selection in strategy but MA can still benefit from EIE to get further improvement.

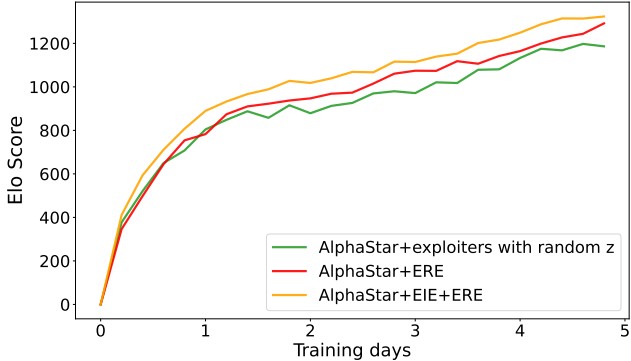

Figure 9: Comparison of the different settings of exploiters.

## C.2 Learning Process of Explorative Exploiters

Explorative Exploiters are designed to learn the strict build orders in $z$, which is especially useful for $z$ that are currently under-explored. In this section, we select a set of $z$ with high execution deviation and compare the learning efficiency of exploiters on these $z$ with various settings. For a specific $z$, we exhibit the learning process of each entity in the sequence by showing the increase in their execution precision. The execution precision of the $n$-th entity refers to the ratio of successful executions of the $n$-th entity after the successful execution of the first $n-1$ entities. We represent each entity in a different color in sequence in the following figures in this section.

Take the strategy Proxy Stargate for example, with its build order as "Gateway->Assimilator->Assimilator->Gateway->Cyberneticscore->Stargate->Adept->Adept->Gateway->Voidray->Shieldbattery->Shieldbattery->Shieldbattery->Voidray->Voidray->Nexus->Voidray". Figure 10 compare the learning efficiency of the exploiter on this strategy with different learning settings. With the help of z-related curricular reward and strategy-guided loss, the agent learns to execute the strategy defined by $z$ effectively and accurately.

In Figure11, we provide the learning process of the build orders on another six $z$ as shown in Table 4 under the setting of Explorative Exploiters. Within $4 * 10^8$ steps, which is consistent with the training steps of ME, all the entities specified by target $z$ achieve an execution precision above 50%. In Table 4, we also provide a specific comparison of the last entity execution precision of the AlphaStar z-related pseudo-rewards setting and the ERE setting.

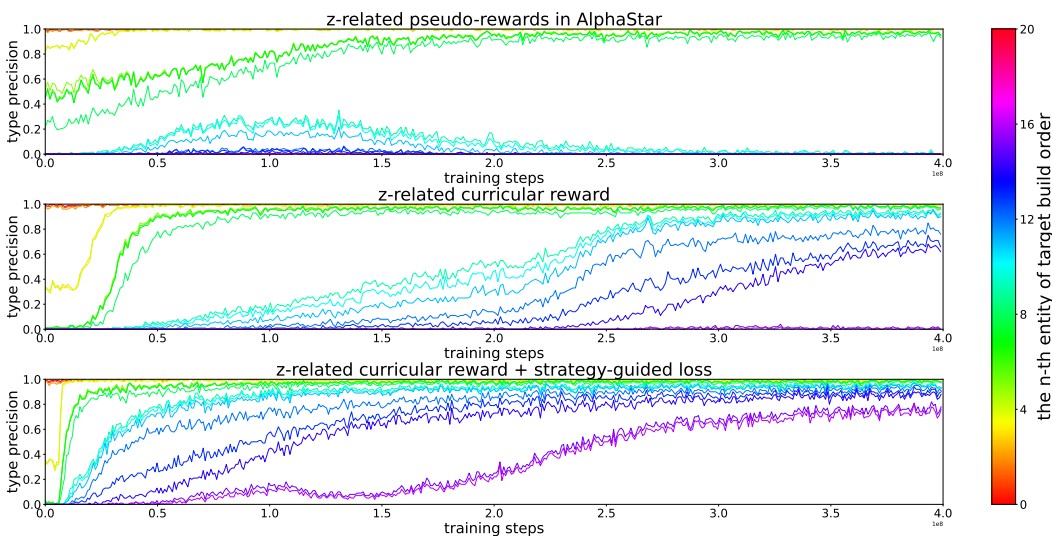

Figure 10: Comparison of the learning process on an under-explored strategy (Proxy Stargate) with different exploiter settings. The top one is the exploiter with the original z-related pseudo-rewards in AlphaStar. The middle one is the exploiter equipped with the z-related curricular reward. The bottom one is the final Explorative Exploiter with the z-related curricular reward and strategy-guided loss.

Table 4: The execution precision of the last entity in target build order after training of $4 * 10^8$ steps. **AlphaStar** refers to training the exploiter with z-related pseudo-rewards as in AlphaStar. **ERE** refers to the exploiter settings in ERE, including the z-related curricular reward and strategy-guided loss.

| index | target build order | last entity execution precison | |
|---|---|---|---|
| | | AlphaStar | ERE |
| 0 | Forge->Assimilator->Gateway->Assimilator->Gateway ->Cyberneticscore->photoncannon->Gateway->Stalker->Stalker ->Stargate->Shieldbattery->Shieldbattery->Shieldbattery ->photoncannon->Stargate->Voidray->Shieldbattery ->Shieldbattery->Voidray | 0 | 0.69 |
| 1 | Gateway->Assimilator->Assimilator->Gateway->Cyberneticscore ->Roboticsfacility->Stalker->Stalker->Warpprism->Roboticsbay ->Stalker->Stalker->Shieldbattery->Nexus->Disruptor->Stalker ->Stalker->Shieldbattery->Disruptor->Stalker | 0 | 0.69 |
| 2 | Gateway->Assimilator->Assimilator->Gateway->Cyberneticscore ->Adept->Adept->Adept->Adept->Nexus->Adept->Adept ->Shieldbattery->Adept->Adept->Roboticsfacility->Sentry ->Sentry->Sentry->Sentry | 0 | 0.80 |
| 3 | Gateway->Assimilator->Assimilator->Gateway->Cyberneticscore ->Twilightcouncil->Adept->Adept->Adept->Adept->Darkshrine ->Gateway->Nexus->Shieldbattery->Darktemplar->Darktemplar ->Roboticsfacility->Stalker->Stalker->Stalker | 0 | 0.50 |
| 4 | Gateway->Assimilator->Assimilator->Gateway->Cyberneticscore ->Adept->Adept->Stargate->Stalker->Stalker->Shieldbattery ->Stalker->Oracle->Stalker->Stalker->Shieldbattery | 0 | 0.60 |
| 5 | Gateway->Assimilator->Assimilator->Gateway->Cyberneticscore ->Adept->Adept->Stargate->Stalker->Stalker->Oracle->Nexus ->Oracle->Roboticsfacility->Twilightcouncil->Stalker->Stalker ->immortal->Stalker->Stalker | 0 | 0.67 |

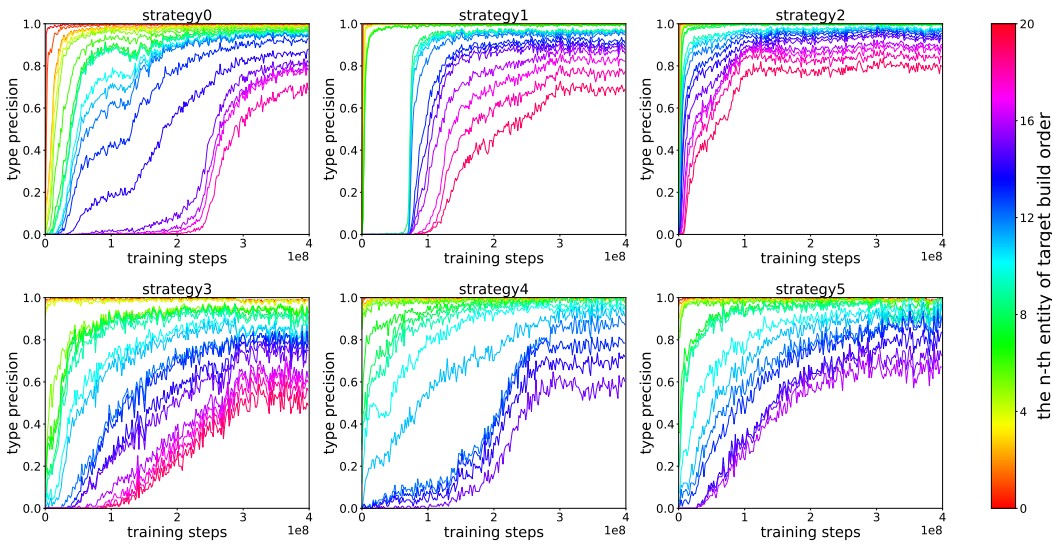

Figure 11: 6 cases of the Explorative Exploiters' learning process on under-explored strategies.

## C.3 The Impact of Opponent Modeling on Scouting Ability

We measure the effectiveness of each scouting behavior with the opponent prediction model and reward the agent accordingly. To show the impact of scouting rewards, we compare the time consumed for the agents to discover the opponent's newly-built buildings. We made two 5-day MA models trained with/without opponent modeling play against each other for 2000 matches. To encounter diverse opponents, both two models randomly pick $z$ from set $\mathcal{D}$ to execute. Then we calculate the average interval from the opponent constructing a new building until the agent discovers it. As shown in Figure 12, the scouting reward remarkably reduces the time for discovering the building of opponent under the fog.

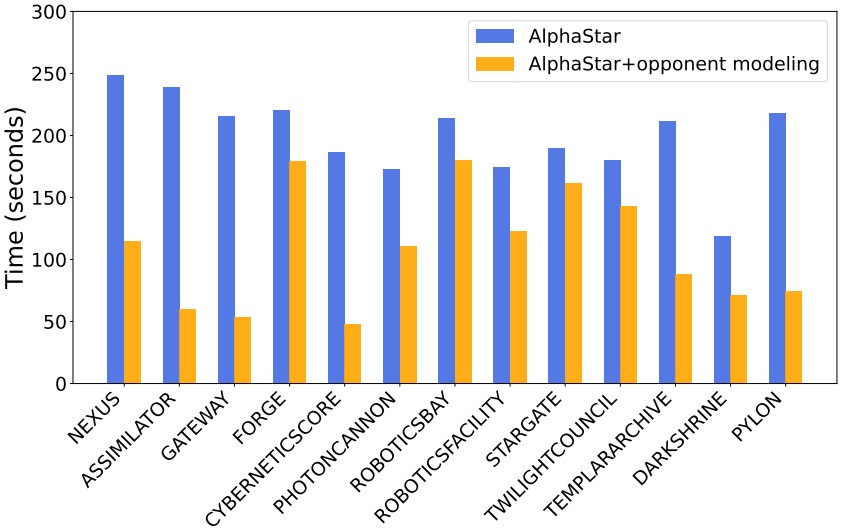

Figure 12: The consumed time to discover opponent's newly-built buildings.

## C.4 Opponent Prediction Model Performance

In this section, we exhibit the performance of opponent prediction model on the test set of 3000 human replays.

We make quantity predictions for the current opponent's entities, including those in production. We categorize the quantity of each entity type into buckets, where buildings are grouped into 0, 1, 2, and greater than 2 categories, and military units are divided into 0, 1, 2, 3-4, and greater than 4 categories. We use multi-classification heads to predict these categories and calculate the macro f1-score of these tasks on the test set, as shown in Figure 13.

Another critical piece of the opponent's strategy is its technical route. Once the performance of the opponent military unit is observed, it is possible to directly infer whether the opponent has upgraded a certain technology, such as "Charge" which can increase the attack speed of Zealots. However, at this time the optimal timing to respond is often missed. Instead, it is more meaningful to predict the technologies under research. As they are binary classification tasks with extremely imbalanced data, we measure the ability of the opponent prediction model with average precision (AP), as shown in Figure 13. The mean average precision (mAP) of prediction on all technologies is 0.73.

The construction location of key buildings, specifically whether they are constructed outside of the base, determines if the opponent is using proxy strategies. Therefore, we utilize binary classification to predict whether the opponent is constructing or has already constructed proxy buildings. The AP of each type is shown in Figure 13 and the mAP of all proxy building types is 0.75.

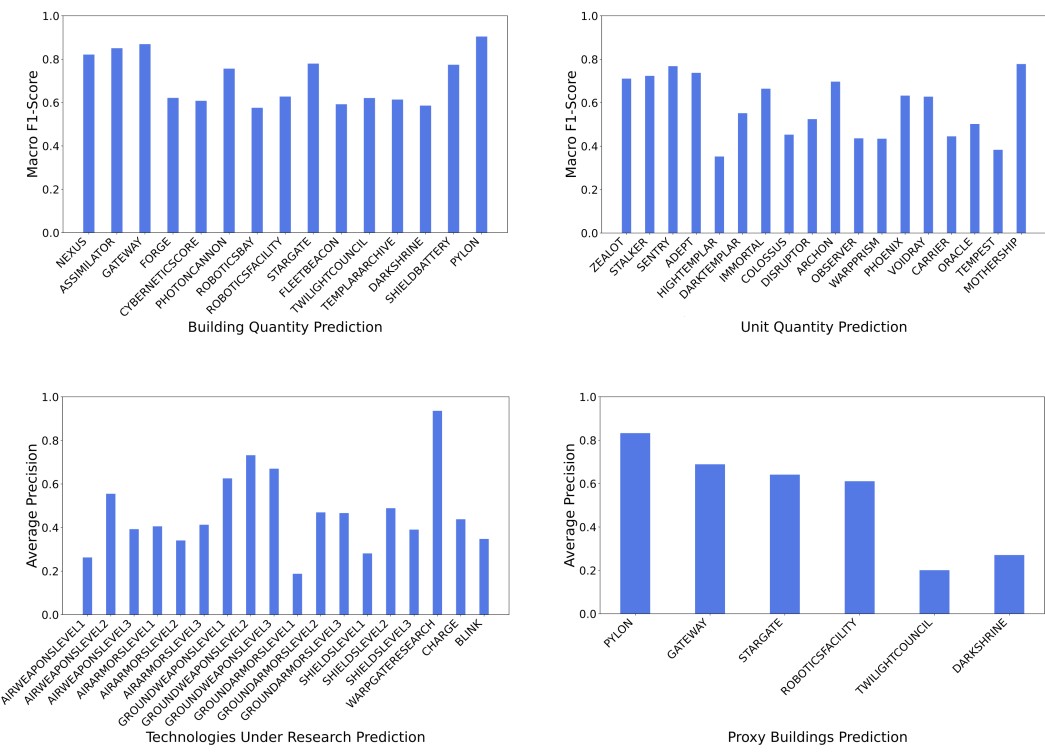

Figure 13: The performance of opponent prediction model on the test set.

## C.5 Strategy Diversity in Exploiters

Intuitively, the robustness of MA would benefit from the strategy diversity in exploiters. In this section, we evaluate the strategy diversity in exploiters of two leagues, ROA-Star and AlphaStar, in terms of both qualitative and quantitative measures.

To get a vectorized description of each model generated by the exploiters during the first 10 days, we can have them play 100 matches against a common opponent, such as the 3-day MA model in

AlphaStar. For each model, we calculate the average statistics of entities and technologies on the matches to generate a vectorized description. Then we analyze their strategies by applying k-means clustering to these vectors. The clustering result, shown in Figure 14, displays each model as a point in the 2d space after their vector dimensions have been reduced using t-SNE. The exploiters in ROA-Star can explore more strategies than AlphaStar within the same training time and resources, e.g. the proxy strategies.

We also quantitatively analyze the diversity in exploiters like Determinantal Point Process (DPP, Kulesza and Taskar 2012) dose. DPP measures the diversity of a candidate set by calculating the determinant of a kernel matrix that describes the similarity between each item pair in the candidate set. To get the similarity matrix, we calculate an L2-distance $d$, then use the kernel function $\exp(-\frac{d}{T})$ to transform the distance into similarity, $T$ is a scale factor which we set to 3 here. The final DPP scores of ROA-Star and AlphaStar are shown in Table 5. The numerical comparison of the DPP support that exploiters in ROA-star are more diverse than AlphaStar.

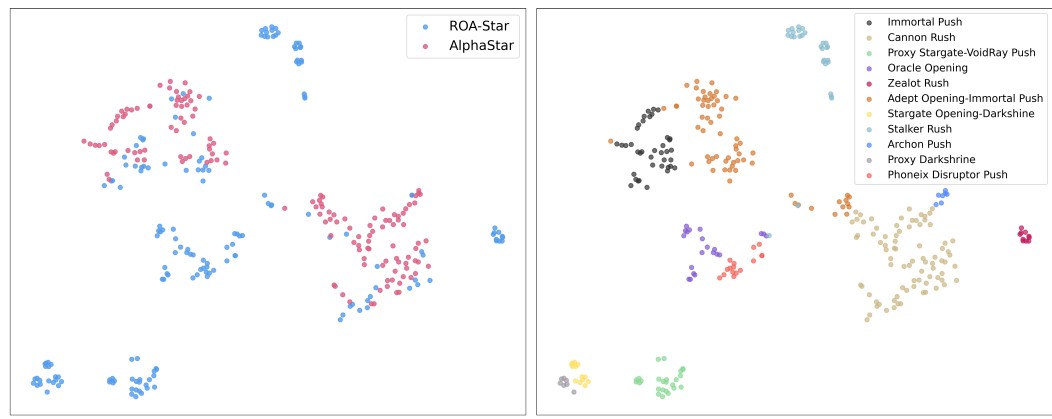

Figure 14: Left visualizes the models in exploiters of AlphaStar and ROA-Star in the 2d space with t-SNE. Right colors the left points with the result of K-means clustering on the models.

Table 5: Diversity in Exploiters

|  | AlphaStar | ROA-Star |
| --- | --- | --- |
| DPP score | 0.002 | **0.229** |

### C.6 Latent Space of Opponent Prediction Model

In this section, we visualize the latent space of the opponent prediction model when the agent fights against four different opponents, as shown in Figure 15(b). From top to bottom, we present the distribution of $h_t$ (the latent opponent strategy embedding) at different game stages: at the 30th second of the game, all the opponents only produce some Probes (the workers) and can't be distinguished. In the 3rd minute, the opponents build the first tech building and choose different tech routes, but they still have similar early military units (like Stalkers), so there are overlaps in the hidden space. In the 6th minute, the opponents produced different units and developed different technologies, resulting in a distinct distribution in the hidden space. As a comparison, Figure 15(a) visualizes the latent space of the policy network in AlphaStar. When facing different opponents, there is no obvious distinguishability in the distribution of hidden states.

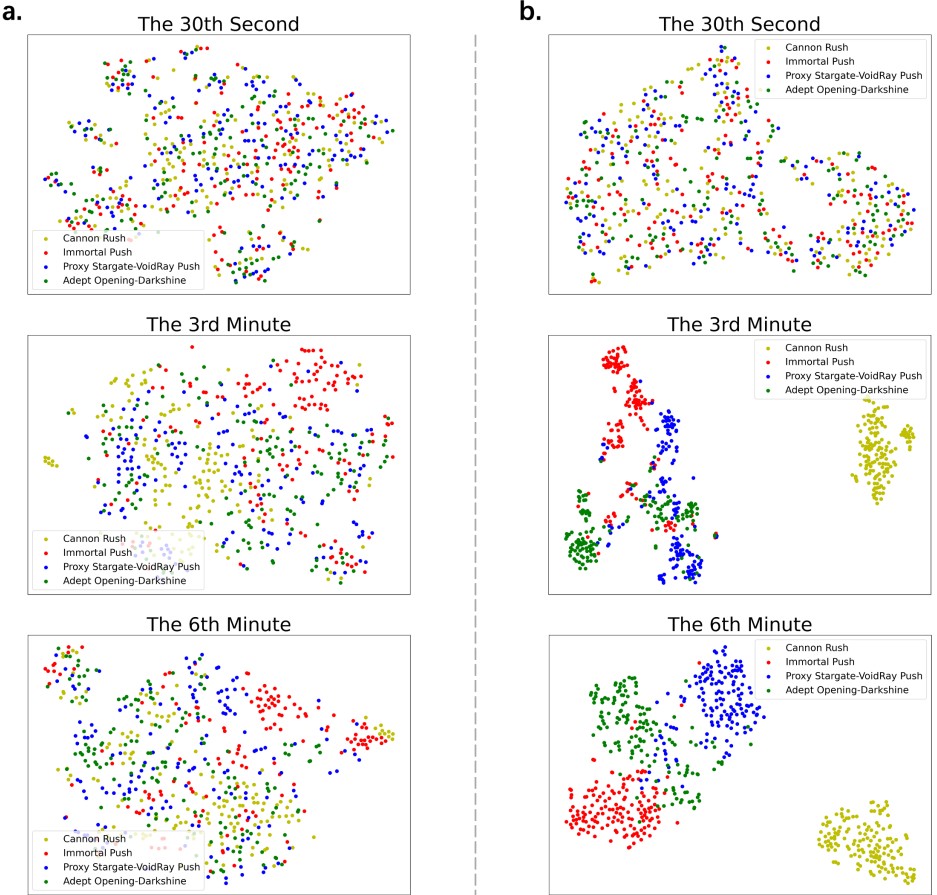

Figure 15: Let AlphaStar and ROA-Star fight against four specific opponents, each for 150 matches, and visualize (a). the hidden states of the LSTM layer in AlphaStar's policy network (b). the latent opponent strategy embedding $h_t$ generated by ROA-Star's opponent prediction model. From top to bottom, we collect the corresponding hidden states at different game stages and show their distributions in the 2d space using t-SNE. We color each sub-figure according to the agent's opponents, which includes 150 points for each opponent.

