# OpenReview forum: "A Robust and Opponent-Aware League Training Method for StarCraft II"
_NeurIPS.cc/2023/Conference — NeurIPS 2023 spotlight_

### Official Review · Reviewer_eEEJ · 2023-06-16

**Soundness:** 3 good
**Presentation:** 3 good
**Contribution:** 2 fair
**Rating:** 6
**Confidence:** 4

**Summary:**

The paper aims to reduce the computational complexity of league training (the AlphaStar system) for Starcraft II by introducing heuristics. Exploiter agents are used in AlphaStar to target weaknesses in the primary policy (main agent) being trained. The paper proposes to condition the exploiter agents on strategies that are either (1) high win rate or (2) divergent from existing strategies. This provides a more guided set of agents that force the main agent to respond to strong opponents or diverse strategies. Another heuristic aims to improve performance by augmenting the base model with a (frozen) opponent prediction model tasked with predicting opponent state. This is used to design a reward that encourages the model to reduce the entropy in opponent model predictions, which creates "scouting" behavior that is beneficial in Starcraft II. Ablations show the components added improve model performance compared to the baseline AlphaStar and a human study against expert players demonstrates (modest) success over a large number of repeated matches.

**Strengths:**

## originality

Modest. Augmenting the league training algorithm with more specific strategy conditioning is a welcome addition to techniques for improving the opponents used in league training in a general fashion.

Opponent modeling is not a new task for game playing, but the specific implementation is new (to my knowledge). A scouting reward is an obvious fit to a game like Starcraft II where opponent knowledge is crucial. I look forward to seeing this more widely tested on other hidden information games where scouting may involve stronger trade-offs, like Honor of Kings (meaning: games where scouting comes at a cost compared to other tasks; in Starcraft II most scouting can be done relatively cheaply and in parallel to other activities).

## quality

Good. The human study is very strong evidence of ecological validity and success. Evaluating against 20 matches is a strong criteria and welcome change compared to the previous best of 3 or 5 scenarios.

The results showing AlphaStar exploiters lose efficacy later in training (figure 6) are another example of solid evaluations of the core algorithms. The main method ablations all reflect a clear attention to evaluating the new methods.


## clarity

Modest. The methods and experiments are all explained in detail. Some of the details were difficult to follow and would benefit from alternative explanations (see below). Figures were mostly easy to understand, though some of the text assumes familiarity with details of Starcraft II (like Figure 4 and Table 2) that many readers may not possess.

## significance

Modest. General league training improvements are of value to RL algorithms applied to large strategic spaces. This will be of interest to a meaningful subset of the RL community investigating game playing. The improvements in saved computational time are modest, but clearly present.

Opponent modeling and the scouting reward are more specific to Starcraft II but will likely apply to certain other games. Overall the methods are promising as a default alternative to the baseline league training methods with no clear downsides aside some code complexity to implement the strategy tracking statistics and selection methods.

**Weaknesses:**

The results show modest improvements, but it is not clear how the long-run scaling looks. Does ROA-Star asymptote to the same performance as AlphaStar? Reach higher? Figure 2 suggests a meaningful Elo score improvement by 5 days of training, but it was not clear if AlphaStar would eventually catch up, and if so what that gap looks like.

The win rates against human experts converge to nearly 50%. Is there evidence that the AlphaStar model trained equivalently would do worse?


**Questions:**

- [Q1] How does ROA-Star scale?
	- How much can the core work be parallelized across more GPUs or agents (to reduce training time)?
	- Are there any particular features of scaling that would differ to AlphaStar?
	- Would ROA-Star benefit more or less from maintaining multiple concurrent exploiters of a given type?
- [Q2] Please provide pseudo-code for the EIE algorithm and ERE algorithm in the main text.
	- The details are a bit murky regarding deviation and how it is used to guide sampling. These are the core new methods being introduced, so they should be very clear from the exposition.
	- Particularly given no code will be released, this will substantially improve reproducibility.
- [Q3] Section 5.2
	- Consider expanding this analysis more for readers unfamiliar with Starcraft II. What is Figure 4 showing?
		- It may help to reorganize Figure 4. Show the two models (AlphaStar with and without opponent modeling) compared for each specific building route. So two colors (with vs without opponent modeling), with a separate box for each strategy transition.
		- Or consider a summary metric that measures the rate of changes in build order in general.
		- Or perhaps move this mostly to the appendix. It may be sufficient to provide a simple metric showing greater knowledge of opponent state without the specific strategic changes made.
	- What are good summary statistics to show tactical changes due to scouting?
	- Overall 5.2 was difficult to follow and it's not immediately obvious that the opponent modeling is changing strategy (correctly) more than the baseline.
- [Q4] Figure 6 is important!
	- This is an important phenomenon to document and emphasize.
	- I've not seen the degraded exploiter performance reported elsewhere. Creating a metric to track and benchmark this performance decay would help others evaluate other league training methods and assess potential improvements.
- [Q5] Figure 7
	- Mark the 50% win rate line clearly. This will make it more obvious how well/poorly the agent does.
	- Please report the final win rates over 20 matches against each human.


**Limitations:**

The paper does not discuss societal impact of the work. This is not that important given the focus on a game that is well discussed - the societal consequences of this work do not alter the conclusions made about the original AlphaStar work.

Limitations are not addressed with specific details. The paper would benefit from pointing out weaknesses in the method or cases that are not currently addressed by the algorithms. Some example questions: When might the scouting reward harm overall performance? What are the limits of a frozen opponent prediction model? Why is performance so close to 50% against human experts?

---

> ### Author Rebuttal · Authors · 2023-08-09
>
> Thanks very much for your review. We respond to all your questions below, and we are happy to provide further details if there is anything still unclear.
>
> **Weaknesses**: It is not clear how the long-run scaling looks? Is there evidence that the AlphaStar model trained equivalently would do worse?
>
> **Answer**:
> We reimplemented AlphaStar and trained it for 10 days in total, where the comparisons with ROA-Star are reported in Section 5.3. Due to limited resources, we stopped training AlphaStar after a thorough evaluation. First, there is an obvious gap in Elo scores between the AlphaStar’ MA and ours (Figure 5). We found that AlphaStar's MA is very fragile when facing certain models in the ROA-Star league, resulting in the worst win rate of 1% (Table 3). Also, we observed a decrease in the effectiveness of its exploiters (Figure 6), and its league training gradually degraded to self-play. We believe that with an extended period of training, it’s unlikely for AlphaStar to catch up with ROA-Star.
>
> For a comparison (see Table 1) between DeepMind's AlphaStar and ROA-Star. Our MA trained for 50 days which consumed about $4.42 * 10^{10}$ steps, and DeepMind's MA trained for 44 days and consumed around $1.9 * 10^{11}$ steps. In repeated games against top humans, our agent maintains a win rate of no less than 50%, which means the top humans didn’t find a winning strategy that can repeatedly defeat our agent. Other than the results shown in Table 1, DeepMind's AlphaStar was easily defeated by Grandmaster players in Battle.net with the Cannon Rush strategy (0 wins and 2 losses, PvP), which is one of the uncommon strategies.
> (DeepMind has released the replays of AlphaStar's matches on Battle.net against human players, here is the link https://www.nature.com/articles/s41586-019-1724-z#Sec32.)
>
> **Q1**: How does ROA-Star scale
>
> **A1.1**:  More GPUs can support us to increase the batch size of agent training, thereby accelerating the training speed. Specifically, we have tried increasing the number of GPUs for training a single agent from 64 to 128. By utilizing data parallelism, we were able to double the batch size. Although there was an increase in communication time between GPUs during gradient backpropagation, the overall efficiency improved by nearly 30%.
>
> **A1.2**: Actors in AlphaStar require both CPUs and TPUs to generate samples, while ROA-Star's actors only require CPUs. ROA-Star can accelerate sample generation speed by increasing the number of CPUs.
>
> **A1.3**: We haven't conducted any relevant experiments, but we can offer some conjectures. Under the framework of ROA-Star, maintaining multiple concurrent exploiters of a given type would find more weaknesses of the main agent and the entire league, which is beneficial to improve the robustness of the main agent.
>
> **Q2**: Please provide pseudo-code for the EIE algorithm and ERE algorithm in the main text.
>
> **A2**: Thank you for the suggestion. We will add the relevant pseudocode. Here, we provide some details on calculating deviation. As we mentioned in Section 2.2, the statistic $z$ includes the first 20 build order (buildings or units) of a game. During training, MA conditions on a certain $z$ sampled from human replays and generate the actual executed build order, which might deviate from the build order in $z$. To measure the deviation in execution, we calculate the edit distance between the actual executed build order and the target build order once the game is over. For each $z$ in $D$, we maintain its moving average of the execution deviation. We will make this clearer in the revision.
>
>
> **Q3**: Section 5.2
>
> **A3**: We have reorganized Figure 4 as you suggested, as shown in Figure 2 in the PDF we submitted.
>
> Given StarCraft II is a highly complex game, it is hard to define a metric to summarize all reasonable strategy transitions. Therefore, we let the agent play against four specific opponents and show its distribution of the first two tech buildings, which is a coarse representation of a strategy.
>
> For readers who are unfamiliar with Starcraft II, we have demonstrated that opponent modeling increases the agent's flexibility in responding to different opponents, as well as improving win rates in all situations, as shown in Table 2; For Starcraft II players, it is easy to check that the strategy transitions depicted in Figure 4 are straightforward and reasonable.
>
> **Q4**: Figure 6
>
> **A4**: In the original AlphaStar paper, Extended Data Fig. 8 plots a payoff matrix of league training, which shows a similar phenomenon: the effectiveness of exploiters degrades as training progresses. Although AlphaStar did not further investigate this phenomenon, we believe that improving the effectiveness of the exploiter throughout the entire training process would be beneficial for the league training.
> Except for the payoff matrix, a moving average of the win rate for each exploiter defeating the MA is a meaningful metric that indicates the effectiveness of the exploiter, and this metric can be easily obtained during the training process. We will make it clearer in the revision.
>
> **Q5**: Figure 7
>
> **A5**: Thank you for your suggestion. We have refined Figure 7 and show the revision in Figure 3 in the PDF we submitted. We will add a table to report the final win rates in the final version.
>
>
> **Limitation**: Limitations are not addressed with specific details.
>
> **Answer**：We are willing to discuss the limitations of the frozen opponent prediction model, and we will incorporate this discussion into the revision.
> The frozen opponent prediction model preserves the prior knowledge of game strategies that human players consider effective. The diversity of strategies in human data may affect the effectiveness of opponent modeling. In other words, if the agents discover new and reasonable strategies during league training, the effectiveness of opponent modeling might decrease. This is something we need to verify in the future work.

---

> > ### Comment · Reviewer_eEEJ · 2023-08-12
> >
> > Thank you for the detailed replies and revised figures.
> >
> > - [A1] Great to see the reasonable scaling benefits! I did not realize actors require TPUs in AlphaStar. Knowing ROA-Star can benefit from CPU-only scaling is another benefit to highlight in the paper.
> > - [A3] This is much clearer than before.
> > - [A5] Much better! As a minor point I would suggest using a dashed or solid black line across the 50% y-value to make the value clearer. Either way this is much easier to read.

---

> > > ### Author Response · Authors · 2023-08-14
> > >
> > > Thank you. We will refine Figure 7 according to your suggestion.

---

### Official Review · Reviewer_zEct · 2023-07-04

**Soundness:** 3 good
**Presentation:** 3 good
**Contribution:** 3 good
**Rating:** 7
**Confidence:** 4

**Summary:**

This paper describes an AlphaStar-like approach for training a top-human-level StarCraft 2 agent, with three core additions/changes to the approach that was used to train AlphaStar:

1. Training exploiters in the league conditioned on certain goals: exploitative exploiters which are conditioned on the $z$ statistics (i.e., unit composition strategies) that perform best for the main agent, and explorative exploiters which are conditioned on under-explored $z$ statistics.
2. An auxiliary loss (used in supervised learning on expert replays phase) to predict what strategy (how many of which unit types) the opponent is using.
3. Reward shaping (used in reinforcement learning phase) that rewards the agent when it improves its own ability (through scouting) to predict what units the opponent has with the part of the model that was trained on the axuiliary loss mentioned above.

The contributions are primarily intended to make the overall agent more robust / less susceptible to unexpected strategies.

**Strengths:**

- Strong empirical results (including against top human experts)
- The writing is clear (barring just a few minor parts) and correct

**Weaknesses:**

### Main comments
- The paper feels seems to barely fit in a conference submission. Results for several experiments (more than just hyperparameter tuning / extensive ablations) are only very briefly mentioned in the main paper (with results not even summarised, just the existence of the experiment being mentioned), with results+discussion only present in supplementary material.
- The paper mentions AlphaStar's computational costs being prohibitive, and it is probably true that this paper used a bit less.... but, frankly, the hardware used for this paper is **still** extremely much.

### Detailed comments
- Table 1 does not mention for how long each agent was trained. This was 50 days for ROA-Star, but (I think?) less for AlphaStar. I'm not sure by how much. I imagine that, even if AlphaStar took much less time, the 256 TPUs would still be more expensive than the 64 GPUs... but anyway, this is important information to include here.
- Line 58: Last but no least --> Last but not least
- Lines 114-115: this is a bit vague / overly general. "does not respond [to] the opponent['s] real-time strategy" --> what does it mean to "respond" to a strategy? Surely the agent plays, it must be responding somehow. What is the difference between "real-time strategy" and any other form of "strategy"?
- Line 123: More specially --> More specifically
- Lines 139, 140, 275: z (without mathmode) --> $z$
- Line 144: In instead --> Instead
- Line 174: "there has been evidence showing that" --> this needs a reference
- Line 288: "Similarly" --> similar to what?

**Questions:**

Can you provide any clarification in particular on the point I described above about Table 1?

**Limitations:**

No comments.

---

> ### Author Rebuttal · Authors · 2023-08-09
>
> Thanks very much for your review. We respond to your questions below.
>
> **Q1**: Can you provide any clarification in particular on the point I described above about Table 1?
>
> **A1**: Yes. The agents of AlphaStar were trained for 44 days. With the scale of our computational resources, an agent can process about 11,000 environment steps per second. After 50 days of training, each of our agents (MA, ME, and 2 LEs) consumed around $4.42 * 10^{10}$ steps.  By comparison, each agent of AlphaStar can process about 50,000 environment steps per second and consumed around $1.9 * 10^{11}$ steps during the entire training process. We will make this clearer in the revision.
>
> **Q2**: Writing issues.
>
> **A2**: Thanks very much for your suggestion. We will edit the paper according to your advice.
>
> **Q3**: Lines 114-115: this is a bit vague / overly general. "does not respond [to] the opponent['s] real-time strategy" --> what does it mean to "respond" to a strategy? Surely the agent plays, it must be responding somehow. What is the difference between "real-time strategy" and any other form of "strategy"?
>
> **A3**:  Responding to a opponent strategy means (in our statement) adjusting one's strategy (by building tech buildings, training military units, and researching new technologies) in order to gain a strategic advantage. For example, if it is discovered that the opponent has produced stealth units (like Dark Templars), one should produce anti-stealth units (like Observers). Opponent real-time strategy refers to the opponent's current strategy, rather than the opponent's historical strategy in early times. We will make this clearer in the revision.
>
> **Q4**: Line 174: "there has been evidence showing that" --> this needs a reference
>
> **A4**: We observed this phenomenon in the human evaluation replays of AlphaStar released by DeepMind ( https://www.nature.com/articles/s41586-019-1724-z#Sec32). It is relatively easy to observe that AlphaStar rarely conducts effective scouting and lacks knowledge of the opponent's real-time strategy, making it fragile to uncommon strategies like the Cannon Rush strategy in PvP matches (as shown in our supplementary video). We will add this reference in the revision.
>
> **Q5**: Line 288: "Similarly" --> similar to what?
>
> **A5**: Similar to the evaluation method used in section 5.1. We will make this clearer in the revision.

---

> > ### Comment · Reviewer_zEct · 2023-08-14
> >
> > Thanks. This message is just to confirm that I have read your rebuttal. I have no further specific questions at this time.

---

> > > ### Author Response · Authors · 2023-08-15
> > >
> > > Many thanks for your thorough review and suggestions.

---

### Official Review · Reviewer_Xpov · 2023-07-05

**Soundness:** 4 excellent
**Presentation:** 3 good
**Contribution:** 4 excellent
**Rating:** 8
**Confidence:** 4

**Summary:**

This paper introduces ROA-Star, an improvement to the AlphaStar training framework for StarCraft II. ROA-Star addresses two identified issues in AlphaStar: diminishing efficiency of exploiters as the training progresses and the Main Agent's inability to adapt to opponent strategies in real time. As a solution to the first problem, it incorporates goal-conditioned reinforcement learning with two types of exploiters - Exploitative Exploiters (EIE) and Explorative Exploiters (ERE). EIE fine-tunes micro-level operations targeting high win-rate strategies, while ERE uncovers macro-level strategy weaknesses. ROA-Star also utilizes opponent modeling through a probabilistic encoder-decoder, and includes a "scouting" reward to encourage agents to actively collect information on the opponent. Experiments demonstrate that ROA-Star enhances robustness, adaptability, and performance compared to AlphaStar.

**Strengths:**

The paper proposes a new approach to league-based training in complex imperfect-information zero-sum games. The authors elect to apply their approach to StarCraft II, which is arguably one of the most complex environments in this domain. They offer clear comparisons against the state-of-the-art AlphaStar solution, demonstrating that their method outperforms the baseline across various metrics, including learning speed, computational resources needed, and robustness of the final agent. Compelling evidence of their approach's superiority is provided through comparisons of their agent against AlphaStar and human players in a more challenging setup than what AlphaStar employed, involving multiple games allowing humans to exploit any weaknesses in the agent. This is a particularly strong paper.

**Weaknesses:**

This paper is a really strong contribution, but it could be further strengthened by refining the writing. Specific examples include:

line 144: The word 'In' is superfluous.
line 171: 'seen' should be replaced with 'observed'.
line 260: 'Besides' should be replaced with 'Additionally'.
line 295: 'ELo' should be corrected to 'Elo'.
lines 113-114-115 need to be rewritten for clarity and coherence.

Addressing these linguistic issues would enhance the paper's readability and presentation.

**Questions:**

Do the authors have any intentions to release the game replays involving matches against human professionals? Additionally, are there plans to make the developed agent accessible to the public? These aspects could offer valuable insights and opportunities for further research and community engagement.

**Limitations:**

The authors acknowledge that their approach relies on certain StarCraft-specific assumptions. It would be really interesting to see how this method performs in different domains, including simpler environments such as Stratego. This extension could provide a more comprehensive understanding of the approach's applicability and limitations beyond StarCraft.

---

> ### Author Rebuttal · Authors · 2023-08-09
>
> Thanks very much for your review. We respond to your questions below.
>
> **Q1** (weaknesses): This paper could be further strengthened by refining the writing.
>
> **A1**:  Thanks very much for your suggestion. We will edit the paper according to your advice.
>
> **Q2** (Questions): Do the authors have any intentions to release the game replays involving matches against human professionals?
>
> **A2**: Thank you for your interest. We have originally included all the game replays against human professionals in the supplementary materials under the folder named 'replay'. We will make this clearer in the paper.
>
> **Q3** (Questions): Are there plans to make the developed agent accessible to the public?
>
> **A3**: We are willing to make it public, but unfortunately, it is currently technically infeasible. Blizzard released StarCraft API in https://github.com/Blizzard/s2client-proto with the highest supported game version of 4.10.0, which is the version we trained our agent on. But the current version of its client has been updated to 5.0.11. It requires configuring the player's local StarCraft II client to be able to play with our agent on an earlier version. For the same reason, we cannot release our agent on Blizzard's online matching system Battle.net like AlphaStar did, as the players play on the latest game version.
>
> **Q4** (limitation): It would be really interesting to see how this method performs in different domains, including simpler environments such as Stratego.
>
> **A4**: One direction of our future work is to apply ROA-Star to other large-scale two-player imperfect-information games. Compared to small games, ROA-Star is more likely to offer superior performance  in large games with high complexity. This is because ROA-Star can be viewed as a trade-off between efficiency and theoretical guarantees. For smaller games (compared to StarCraft II), there have been extensive methods with strong theoretical convergence guarantee to Nash Equilibrium, e.g., PSRO, DeepNash, CFR, etc.

---

### Official Review · Reviewer_sqN1 · 2023-07-07

**Soundness:** 4 excellent
**Presentation:** 4 excellent
**Contribution:** 4 excellent
**Rating:** 7
**Confidence:** 4

**Summary:**

The authors present a modification to the AlphaStar training algorithm that increases robustness and allows for faster responses to opponent behaviour by explicitly encoding the opponent's strategy in the model's representation along with adding scouting as an ancillary objective.

**Strengths:**

The method exceeds the SOTA in starcraft and should generalize to other tasks
They present ablation that demonstrate the individual improvements of each contribution


**Weaknesses:**

There are limited implementation details provided and the authors do not provide a code release.
The authors describe the model as "opponent aware", but this is a hypothesis, they do not investigate if the features they add are adding to "awareness" just if they improve the winrate, the models could just be learning a more robust set of strategies.
Adding scouting as an additional reward objective appears not to have any grounding beyond the observation that the previous models did not have a strong enough bias towards scouting

**Questions:**

Can you provide summary statistics or a full release of the self-play training data? 50 days does not clearly explain how many steps/games were needed to train each model
Did the authors attempt to examine the opponent model latent space?

**Limitations:**

The use of human subjects without IRB is concerning

---

> ### Author Rebuttal · Authors · 2023-08-09
>
> Thanks very much for your review. We respond to your questions in the following.
>
> **Q1**: Did the authors attempt to examine the opponent model latent space?
>
> **A1**: Thank you for your suggestion. To show the agent's 'awareness' of the opponent's strategy, we add an experiment to visualize the latent space of the opponent prediction model when the agent fights against four different opponents, as shown in Figure 1.b in the PDF we submitted. From left to right, we offer the distribution of $h_t$ (the latent opponent strategy embedding) at different game stages: at the 30th second of the game, all the opponents only produce some Probes (the workers) and can’t be distinguished. In the 3rd minute, the opponents build the first tech building and choose different tech routes, but they still have similar early military units (like Stalkers), so there are overlaps in the hidden space. In the 6th minute, the opponents produced different units and developed different technologies, resulting in a distinct distribution in the hidden space. As a comparison, Figure 1.a visualizes the latent space of the policy network in AlphaStar. When facing different opponents, there is no obvious distinguishability in the distribution of hidden states.
>
> **Q2**: Adding scouting as an additional reward objective appears not to have any grounding beyond the observation that the previous models did not have a strong enough bias towards scouting.
>
> **A2**: Scouting is essential in RTS games because of the existence of “fog of war”. A consensus among StarCraft II human players is that scouting is indispensable to detect the opponent's real-time strategy and enable proper strategic responses, which has a large impact on the prob of winning a game. Also, effective scouting is not easy: it is important to decide when and where to scout, which is difficult to write hard rules for. We design an effective and general-purpose scouting reward based on the decrease in cross-entropy between the true opponent strategy and the opponent model's predictions. We believe our scouting reward scheme can be easily applied to similar partially observable games. We will make this clearer in the revision.
>
> **Q3**: Can you provide summary statistics or a full release of the self-play training data? 50 days does not clearly explain how many steps/games were needed to train each model.
>
> **A3**: Yes, here we present the summary of the training process of our agents, where the basic settings of each agent can be found in appendix A.3. With the scale of our computational resources, an agent can process about 11,000 environment steps per second. Our main agent was continuously trained for 50 days, which consumed about $4.42 * 10^{10}$ steps. In comparison, AlphaStar's main agent was continuously trained for 44 days and consumed around $1.9 * 10^{11}$ steps (around 50,000 environment steps per second). During training, our main agent added a frozen copy to the league every $2 * 10^8$ steps, resulting in a total of 221 models. Our ME reset its parameters after $4 * 10^8$ steps at most, which generated 113 models in total. Our LE reset its parameters after $2 * 10^8$ steps at most, and the two concurrent-training LEs generated 216 and 218 models respectively. These details will be added to the revision.

---

> > ### Comment · Reviewer_sqN1 · 2023-08-14
> >
> > Thank you for putting in the time to run the visualization, think it helps strengthen your explanation of the model's performance.
> >
> > I agree that scouting is important in RTS games, and that improving scouting is a noteworthy result in and of itself. I hope that your revisions make it clear that this was the motivation of your changes to the original algorithm.
> >
> > The additional details will be helpful in further extensions of your work, thank you for putting in the time to collect them.
> >
> > The responses have not changed my position, and I have no followup questions.

---

> > > ### Author Response · Authors · 2023-08-15
> > >
> > > It is our pleasure, and thanks very much for your constructive and thorough review. We will refine the paper according to your suggestions.

---

### Author Rebuttal · Authors · 2023-08-09

We have submitted a PDF that includes three figures. In response to reviewer sqN1's question, we demonstrate the agent's 'awareness' of the opponent's strategy in Figure 1 (in the new pdf). Additionally, based on the suggestions from reviewer eEEJ, we have refined the original Figure 4 and Figure 7, and plotted Figure 2 (in the new pdf) and Figure 3 (in the new pdf).

---

### Decision · Program_Chairs · 2023-09-21

**Decision:**

Accept (spotlight)

**Comment:**

This paper proposes an improved version of AlphaStar for the challenging StarCraft II. The idea is to 1) train goal-conditioned exploiters as opposed to unconditioned exploiters in AlphaStar, 2) train agents to predict the opponent's strategy as an auxiliary task, and 3) give scouting reward to encourage information gathering. The proposed method not only outperformed their replication of AlphaStar in terms of training efficiency but also achieved > 50% win rates against top professional players. All of the reviewers unanimously agreed that the proposed method is interesting and sensible, and the result against top professional players is quite significant, considering how complex and challenging StarCraft II is. Thus, I think this work deserves to be presented at NeurIPS.